# Enhancement of Neuroglial Extracellular Matrix Formation and Physiological Activity of Dopaminergic Neural Cocultures by Macromolecular Crowding

**DOI:** 10.3390/cells11142131

**Published:** 2022-07-06

**Authors:** Andy N. Vo, Srikanya Kundu, Caroline Strong, Olive Jung, Emily Lee, Min Jae Song, Molly E. Boutin, Michael Raghunath, Marc Ferrer

**Affiliations:** 1National Center for Advancing Translational Sciences (NCATS), National Institutes of Health (NIH), 9800 Medical Center Drive, Rockville, MD 20850, USA; andy.vo@virginia.edu (A.N.V.); srikanya.kundu@nih.gov (S.K.); ces09f@my.fsu.edu (C.S.); olive.jung@nih.gov (O.J.); emily.lee@nih.gov (E.L.); minjae.song@nih.gov (M.J.S.); mollyboutin@gmail.com (M.E.B.); 2Department of Life Sciences and Facility Management, Institute for Chemistry and Biotechnology (ICBT), Zurich University of Applied Sciences (ZHAW), 8820 Wädenswil, Switzerland; ragh@zhaw.ch

**Keywords:** extracellular matrices, macromolecular crowding, human iPSC derived astrocyte and dopaminergic neurons, drug testing

## Abstract

The neuroglial extracellular matrix (ECM) provides critical support and physiological cues for the proper growth, differentiation, and function of neuronal cells in the brain. However, in most in vitro settings that study neural physiology, cells are grown as monolayers on stiff surfaces that maximize adhesion and proliferation, and, therefore, they lack the physiological cues that ECM in native neuronal tissues provides. Macromolecular crowding (MMC) is a biophysical phenomenon based on the principle of excluded volume that can be harnessed to induce native ECM deposition by cells in culture. Here, we show that MMC using two species of Ficoll with vitamin C supplementation significantly boosts deposition of relevant brain ECM by cultured human astrocytes. Dopaminergic neurons cocultured on this astrocyte–ECM bed prepared under MMC treatment showed longer and denser neuronal extensions, a higher number of pre ad post synaptic contacts, and increased physiological activity, as evidenced by higher frequency calcium oscillation, compared to standard coculture conditions. When the pharmacological activity of various compounds was tested on MMC-treated cocultures, their responses were enhanced, and for apomorphine, a D2-receptor agonist, it was inverted in comparison to control cell culture conditions, thus emulating responses observed in in vivo settings. These results indicate that macromolecular crowding can harness the ECM-building potential of human astrocytes in vitro forming an ultra-flat 3D microenvironment that makes neural cultures more physiological and pharmacological relevant.

## 1. Introduction

The lack of physiologically relevant and predictive preclinical models has limited the success rate for drug approvals [1,2]. For most in vitro cellular models, cells are grown as two-dimensional (2D) monolayers which lack the physiological presence of cues from the microenvironment coming from parenchymal and systemic cell–cell and cell–extracellular matrix (ECM) interactions [3,4]. The ECM not only influences the mechanical stress experienced by the cells but also modulates their access to biological and chemical cues, such as growth factors and ions, thus regulating cell growth, migration, differentiation, and specific function [5]. In the brain, the ECM provides a microenvironment that regulates axonal guidance, synaptogenesis, and the maintenance of synaptic stability and repair following injury [6,7,8]. Both neurons and astrocytes contribute to the neural ECM [9,10], and the composition of the ECM depends on the developmental stage and region of the brain [11]. In late development, the perineuronal nets (PNNs)—composed of hyaluronic acid complexed with chondroitin sulfate proteoglycans (CSPGs), such as aggrecan, veriscan, neurocan, and brevican, and with the glycoprotein tenascin-R—are responsible for synaptic stability [12,13]. In contrast, the basement membrane surrounding cerebral blood vessels is composed of collagen type IV, laminin, fibronectin, and heparan sulfate proteoglycans (HSPGs) [14,15]. The importance of native ECM in neuronal network activity has been demonstrated in several in vivo studies: Recent animal work has shown that malformation of perineuronal nets (PNNs)-like lattice structures of CSPGs at sites of injury in the central nervous system (CNS) restricted developmental plasticity and modulated synaptic properties by enwrapping the inhibitory interneurons [16,17]. Other studies have shown that stress during development and in adulthood reduces neural plasticity by modulating structural-ECM deposition [18,19]. In vivo functional studies using knockout mice have shown that the ECM composition also regulates the release of extracellular proteolytic factors such as thrombin, tissue plasminogen activator, neurotrypsin, neuropsin, matrix metalloproteinases, and MMP-9; it has also been shown to modulate brain activity and be involved in neurological disorders [20,21].

It would be desirable to recapitulate the native brain tissue ECM in neural in vitro cellular systems to ensure their physiological relevance and activity [22]. However, spontaneous cellular ECM deposition in physiological amounts and composition is difficult to achieve in standard aqueous culture conditions [23]. The exogenous addition of isolated ECM components has been used to increase the activity of cells in in vitro neural cultures [24,25]. For example, laminin coatings have been shown to increase neurite extension and increase in the complexity of neurites formed by neuronal stem cells, and more complex fetal brain extracellular matrices increased neuronal network formation in 3D bioengineered model of cortical brain tissue [25]. In native tissues, the production, secretion, and deposition of ECM by certain cell types is a process regulated by mechanical, chemical, and physical cues from the microenvironment [26]. Macromolecular crowding (MMC) is both a biophysical effect and a novel tissue engineering technology that is based on the principle of excluded volume created by large macromolecules [23,27]. In native tissues, endogenous macromolecules in the extracellular environment create steric constraints that limit the space in which relevant biomolecules and other substances can be exchanged and diffuse in and out of the cellular surroundings. These steric crowding effects increase reaction rates and diffusion within the culture system [28], and for matrix-producing cells, they induce ECM deposition [27]. In the past decade, there has been growing interest in utilizing macromolecular crowding in in vitro cell cultures to improve their physiological relevance by inducing the disposition of native ECM secreted with large macromolecules. Several macromolecules have been shown to act as crowders, thereby inducing ECM deposition, including dextran sulfate (DxS), Ficoll 70 and 400 kDa (Fc 70/400), polyethylene glycol (PEG), and polystyrene sulphonate (PSS) [29]. Until now, the in vitro effects of MMC in inducing the cellular production and deposition of ECM in 2D monolayers cell cultures have been demonstrated in a variety of mesenchymal cells, namely fibroblasts, mesenchymal stromal cells, tenocytes [23], and epithelia (keratinocytes) [30].

There is a critical need to create in vitro neural cellular models that faithfully recapitulate the physiology, pathology, and pharmacological of the human CNS to study human neurological diseases and use as assay platforms for therapeutics discovery and development. Toward this goal, we created a functional neural coculture cellular model with human-induced pluripotent stem cell (hiPSC)-derived astrocytes (iAstros) and dopaminergic neurons (iDopas) to recapitulate human cellular physiology with as much fidelity and relevance as possible [31], given that human primary neurons cannot be readily procured and cell lines do not faithfully recapitulate tissue physiology. We chose to create a culture with iDopas, with the long-term goal of developing neural models to study addiction circuitry and neurodegenerative diseases such as Parkinson’s. We demonstrate that MMC can induce secretion and deposition of neural native-like ECM components, including collage IV, laminin, and fibronectin, in cultures of iAstros. Furthermore, when a neural coculture system is created by seeding iDopas on top of the MMC treated astrocyte–ECM “bed” and incubating for two weeks, there was enhanced neuronal physiological activity, including increased synaptogenesis, complex neurites network, and calcium activity. The pharmacological responses of the neural activity of the coculture model were assessed by using metoclopramide, apomorphine, and morphine. Metoclopramide is a dopamine D2 receptor antagonist, apomorphine is a dopamine D2 receptor agonist, and morphine is a μ-opioid receptor agonist. In vivo, D2 receptors are inhibitory, and when expressed in areas with high numbers of dopaminergic neurons (e.g., ventral tegmental area or substantia nigra), they can function as autoregulators of dopamine release [32,33,34]. Thus, antagonism of the dopaminergic receptor with metoclopramide should produce an increase in dopamine release and neural activity, and agonism of this receptor with apomorphine would have the opposite effect, reducing neural activity. Activation of the μ-opioid receptor in dopaminergic neurons in the ventral tegmental area has been shown to increase neuron activity in rats [35]. Our data show that pharmacological responses to compounds targeting receptors expressed on hiPSC-derived dopaminergic neurons in the MMC-treated coculture emulated those expected from in vivo data.

## 2. Materials and Method

### 2.1. Astrocyte Culture

Astrocytes were cultured in 96-well flat-bottom plates (CellCarrier-96 Ultra, PerkinElmer, Waltham, MA, USA). Wells were first coated with 100 µL/well of 0.083 mg/mL of Growth Factor Reduced Matrigel (Corning, Corning, NY, USA, product no. 354230) in ice-cold DMEM (Gibco, Waltham, MA, USA, catalog no. 10569-010) and incubated overnight, at 4 °C. Human-induced pluripotent stem cell (iPSC)-derived astrocytes (“iAstros”) (iCell Astrocytes Kit, 01434 Cat#R1092, Lot#104345, 105136, 105152, 105337, Hanam, Korea) were obtained commercially from Fujifilm Cellular Dynamic International (CDI-Fujifilm, Madison, WI, USA) and cultured according to vendor’s protocol. Briefly, iAstros were gently thawed into astrocyte culture medium comprising DMEM, high glucose, GlutaMAX™ Supplement, pyruvate (Gibco, 10569-010) supplemented with 10% fetal bovine serum (Gibco, catalog no. 16000), and 1X N2 supplement (ThermoFisher, Waltham, MA, USA, catalog no. 17502). The iAstros were then spun at 380× *g* for 5 min, and the viable cells were counted with Countess II (ThermoFisher Scientific, Waltham, MA, USA), an automated cell counter using trypan blue exclusion dye. The Matrigel solution was aspirated from the wells, and 200 µL of iAstros cell suspension in astrocyte culture medium, with at least 95% viability, was seeded in a precoated well of the 96-well plate (13.3 × 10^3^ cells per well, effectively 4.0 × 10^4^ cells/cm^2^). The iAstros cell cultures were maintained at 37 °C, under 5% CO_2_, in astrocyte culture medium, with 100% medium change every other day, for 7 days. We considered the day of iAstro seeding as day 1 on our experimental timeline. All experiments were repeated 3 times, in separate days, from different cell vials for independent biological replicates, and each biological replicate included 3 wells for each treatment group, for technical replicates. For each biological replicate, two plates were seeded with astrocytes. For all biological replication we started with duplicate plates: one of the plates was fixed at day 8 for immunostaining to assess deposition of extracellular matrices, and the second plate was further processed with the astrocyte–neuronal coculture experiments.

### 2.2. Macromolecular Crowding (MMC) Treatment

Twenty-four hours after seeding the iAstros, the cultures were treated with Ficoll70 and Ficoll400 (MMC), L-ascorbic acid (Vit C), or both (MMC + Vit C) (see Figure 1). The MMC treatment was a cocktail of 37.5 mg/mL Ficoll PM70 (Millipore-Sigma, Burlington, MA, USA, catalog no. GE17-0310-50) and 25 mg/mL Ficoll PM400 (Millipore-Sigma, catalog no. GE17-0300-10) prepared in astrocyte culture media. The solution was then filtered through a 0.22 μm pore–size syringe filter. For Vit C treatment, a 100 mM stock solution of L-ascorbic acid (Sigma, St. Louis, MO, USA, catalog no. A8960) was prepared in ultrapure water and diluted 1:1000 into astrocyte culture media (100 μM). For the combined treatment group, the stock solution of L-ascorbic acid was diluted 1:1000 into the filtered MMC cocktail. Treatments were introduced following the complete removal of the astrocyte culture media. The control group received fresh astrocyte culture media without MMC or Vit C. Media exchanges were performed every other day for an additional 6 days. At day 8, cultures were either fixed and stained for extracellular matrix deposition, or media were aspirated, and neurons were seeded on top of astrocytes (see Figure 1).

### 2.3. Astrocyte–Neuron Coculture

Coculture of astrocytes with neurons started on day 8 and continued for 14 days, until day 22. Human-iPSC-derived dopaminergic neurons (iDopas) were procured from Fujifilm CDI (Madison, WI, USA, iCell DopaNeurons, 01279, Cat#R1032, C1028; Lot#102224, 105288, 102614) and seeded on iAstro–ECM bed. Briefly, iDopas were gently thawed into CDI-recommended dopaminergic culture medium (“iDopa media”) consisting of iCell Neural Base Medium 1 (Fujifilm CDI, catalog no. M1010) supplemented with 2% iCell Neural Supplement B (Fujifilm CDI, catalog no. M1029) and 1% iCell Nervous System Supplement (Fujifilm CDI, catalog no. M1031). Cells were spun down at 380× *g* for 5 min and counted with a Countess II an automated cell counter (ThermoFisher, Waltham, MA, USA), using trypan blue exclusion for viability. Because of variable viability between vials, the iDopas were processed with a dead-cell removal kit (Miltenyi Biotec, Bergisch Gladbach, Germany, order no. 130-090-101). Briefly, cells were spun again at 380× *g* for 5 min, the medium was aspirated completely, and the cells were resuspended in Dead Cell Removal MicroBead solution in 1X Binding buffer (from the stock of 20X Binding buffer in the Kit) in sterile double-distilled water (Fisher Scientific, Hampton, VA, USA, NH catalog no. 10977015). Then 100 µL of magnetic MicroBead solution was used per 10^7^ total cells. The cell suspension was incubated with the beads for 15 min at room temperature, and then additional Binding buffer (1X) was added to make the final loading volume 500 µL. The cell suspension was then loaded into a MACS Column on MACS magnetic Separator from Miltenyi Biotec and allowed to pass through the column. The column was then thoroughly washed with 500 µL of Binding buffer (1X) 5 times. The flow-through live-cells suspension was then resuspended in iDopa media and counted again for viability. At least 85% of the viable iDopas were then seeded as 66 × 10^3^ cells per well (2 × 10^5^ cells/cm^2^) in 200 μL of iDopa media on top of the iAstro–ECM bed after carefully removing the media within the well. The plates were incubated at 37 °C with 5% CO_2_. Two days after neuronal seeding, at experiment day 10, the iDopa media were 100% replaced with BrainPhys neuronal medium (StemCell Technologies, Vancouver, BC, Canada, catalog no. 05790) supplemented with 1X N2 (ThermoFisher, catalog no. 175020-01), 1X B27 (ThermoFisher, Catalog no. 17504), 20 ng/mL of recombinant human BDNF (StemCell Technologies, No. 78005), 20 ng/mL of GDNF (StemCell Technologies, No. 78058), 1 μg/mL laminin (Gibco, 23017), 200 nM ascorbic acid (Tocris, Bristol, UK, catalog no. 4055), and 1 mM cAMP (Tocris, catalog no. 1141), designated “BrainPhys + Bardy supp” media [31], to improve neuronal activity. The neural culture was maintained for 2 weeks, with twice-weekly media exchanges (50%) of “BrainPhys + Bardy supp” media. For the study of functional neuronal network activity, the cocultures were transduced with pAAV9.CAG.GCaMP6f.WPRE.SV40 (Addgene, Watertown, MA, USA, catalog no. 100844) at 10^6^ MOI (Multiplicity of Infection) at day 10, at the same time that media were changed to “BrainPhys + Bardy supp” media. Cultures were incubated for 2 weeks to allow the GCaMP6f protein to be expressed.

### 2.4. Immunocytochemistry

Cell cultures were fixed with 4% paraformaldehyde, made up from 32% stock solution (Electron Microscopy Sciences, Hatfield, PA, USA, catalog no. 15714-S), in PBS for 30 min, at room temperature. Samples were then washed with PBS three times and blocked at room temperature on a shaker, for 1 h, with 5% normal goat serum (ThermoFisher Scientific, catalog no. 50197Z) and 0.2% bovine serum albumin (Sigma, catalog no. A9418). Then 0.2% Triton-X (Sigma, catalog no. X-100) was added to the samples that needed permeabilization for intracellular staining. Triton-X was excluded when staining for extracellular matrix. Primary antibodies were diluted in PBS (see Table 1 for dilutions) and added, and samples were incubated at 4 °C on a shaker, overnight. Samples were then washed with PBS three times. Secondary antibodies were added (see Table 1 for dilutions), and samples were incubated at room temperature, on a shaker, for 1 h. Samples were then washed with PBS three times. After the second wash, Hoechst 33342 (Abcam, Cambridge, UK, catalog no. ab228551) was added in 1:2000 dilution and incubated for 15 min, at room temperature, on a shaker. The staining process was completed with one more round of PBS wash. Samples were then imaged on the Opera Phenix (PerkinElmer, Waltham, MA, USA) under 20X water objective, on confocal mode, with the respective filter setting according to the secondary antibody fluorescence.

### 2.5. Image Analysis of Neural Extensions and Branching

The image analysis and quantification of immunofluorescence signal from neural coculture were performed by using the software Harmony v5 (PerkinElmer, Waltham, MA, USA). For live cells’ image analysis, we used the GCaMP6f FITC signal to identify neuronal cells because we did not include a live nuclear marker, and for fixed cell image analysis, we used Hoechst 33342 fluorescence for identifying the nuclei. We used the “Find nuclei: Method B” algorithm, using an area of >30 µm^2^ with common threshold 0.4, for locating nuclei. GCaMP6f and TH primary antibody staining were used again for live and fixed dopaminergic neuronal extensions’ identification, respectively. Then we applied the “Find Neurites with CSIRO Neurites analysis 2” algorithm to mark the neurites with neurites segment from the preselected nuclei. We calculated the max neurites’ length per well/network, total neurites length per well/network, and number of neurites branching per well/network. The data from 3 wells from each MMC treatment group and 3 biological replicates were aggregated for each treatment condition and plotted as bar graphs for comparison.

### 2.6. Live Presynaptic Vesicle Tracking Assay

The BioTracker 510 Green C4 (FM1-43) Synaptic Dye (Sigma-Aldrich, St. Louis, MO, USA, SCT126) was used to measure newly formed neurotransmitter vesicles at the presynaptic ends of live iDopa + iAstros cocultures. The dye’s final concentration was of 4 µM in 50 mM Tyrode solution, and 200 µL dye solution was added per well after removing the media. The cultures were then incubated for 5 min, at room temperature, and imaged with a Phenix HCS reader with a 10X confocal objective, using an FITC filter (λ_Ex_ = 488 nm and λ_Em_ = 522 nm). The raw fluorescent images were analyzed with ImageJ from 5 field-of-views per well, and data from 3 wells per group from 3 biological replicates were normalized to the respective background florescence (dF/F_0_) for that plate and then plotted as bar plots to compare the amount of newly formed vesicles at the synaptic end as a measure of presynaptic plasticity.

### 2.7. Single-Cell Calcium Flux Imaging of Transfected GCaMP6f Neurons and Astrocytes Cocultures

After 2 weeks of incubation of pAAV9.CAG.GCaMP6f.WPRE.SV40 transduction, the GFP signal was measured from iDopas/iAstros cocultures by using a Phenix Opera HCS (PerkinElmer, Waltham, MA) with a 10X water objective. First, the green laser (λ_Ex_ = 488 nm and λ_Em_ = 522 nm) was used to assess the fluorescence intensity of expressed GCaMP6f with a 60 ms exposure time and 50% laser intensity. To measure the basal live calcium flux over time, 300 frames of time-lapsed images were acquired over a 180 s period with 0.6 s intervals (1.6 frames/s). After basal GCaMP6f signal acquisition, drugs were added to culture wells and incubated for 5 min, and the same acquisition process as above was used to acquire perturbations on the calcium flux.

The analysis of single cellular calcium fluorescence dynamics was as follows: The time-lapse images were stacked to provide maximum intensity projection, using the Harmony v5.1 software (PerkinElmer), and transferred to the ImageJ image-processing software. Then ROIs were automatically selected by centering on the neuronal soma by the “LC-Pro” plug-in in ImageJ by providing the ROI diameter of 30 pixels, frame rate of 1.6 s, and an intensity cutoff threshold with a *p*-value of 0.05. Finally, the raw intensity values of the calcium signal were extracted from selected ROIs. For the analysis, each ROI’s data were transferred to “Origin-Pro 9.0” (OriginLab). Single neuronal dynamics were quantified by using the calcium signal peak frequency and peak amplitude. The peaks were detected over time by “positive maximum intensity peak finding method”, using the “batch peak processing algorithm”, with second derivatives of individual ROIs’ raw fluorescence intensities (RFUs) as a change in fluorescence intensity from the initial (dF) [32]. To improve the signal-to-noise ratio, the second derivatives of the RFUs were used after thresholding the baseline. The initial fluorescence level (F_0_) for each ROI was calculated by averaging the RFUs over full acquisition length, and then the amplitude was normalized to dF/F_0_ for each respective ROIs. The peak frequencies as peak per sec (pps), mean peak amplitudes (dF/F_0_), and mean peak width (msec) from each ROIs were calculated. —Minimum of 3 ROIs per well from 3 technically replicated wells per group and 3 biologically replicated plates per experiment were plotted as bar graphs. We repeated the same analysis for each drugs treatment group.

The F/F_0_ values over time for each ROI/per well were exported as a .csv file from the ImageJ to Python for further peak-detection analysis and generation of a correlation heat map. The code for the analysis was obtained from GitHub [(URL: gist.github.com/25086 accessed on 15 March 2021). The data-processing steps were as follows. The calcium fluorescence data (F/F_0_) for each ROI were normalized with a minimum threshold of 1. The mean signal plus standard deviation with 95% confidence was generated over the time-lapsed interval. The output graphs created by the freely downloadable Matplotlib package of Python are available as (1) heatmaps representing calcium fluorescence signal frequency (with corresponding ascending amplitude in blue to yellow scale) across each ROI over time; (2) a correlation matrix plotted as correlation coefficients across all ROIs, representing the synchronicity of calcium fluorescence signal oscillation within each well’s neural network; and (3) the “Synchrony/correlation score” that the code delivered as a quantification of average correlation coefficient across the matrix. Using the LC-Pro identifiers, we randomly chose 12 ROIs with maximum fluctuation over time per well for the correlation analysis. The required cutoff of randomly sampling correlation score was preset to be within 5% of the correlation score of the overall population of ROIs. The number was plotted as bar plot in GraphPad Prism 9 in order to obtain an average correlation factor across the technical and biological replicates.

### 2.8. Pharmacological Perturbation of Calcium Dynamics in Neural Coculture Systems

We assessed the effects of three drugs targeting dopamine and µ-opioid receptors on the calcium dynamics of the iDopas + iAstros coculture system after MMC treatment: apomorphine, a D2-receptor agonist; metoclopramide, a D2-receptor antagonist; and morphine, the µ-opioid receptor agonist. Each compound, except for morphine, was tested at 10 µM. The final concentration of morphine was 15 µM. Each compound was tested in three separate biological replicates, each with three technical replicates. First, 10 mM solutions of the compounds in DMSO were diluted 1:100 in 1X PBS, and then 20 µL (or 30 µL for morphine solution) of freshly made compound solution was added to the wells, after removing an equal volume of medium (final volume in the well remained 200 µL). For the no-compound control, an equal final percentage of DMSO was added as a vehicle control. Drug treatments were performed for 5 min before imaging.

### 2.9. Statistical Analysis

Each experiment was conducted in at least 3 independent biological replicates performed on different days and starting with different vials of cells, and each MMC treatment group or drug treatment group was performed in 3 technical replicates (3 wells) per plate, for a total of 9 wells of data per condition (3 biological replicates × 3 technical replicates), and these were pooled and analyzed collectively. Bar graphs were plotted in GraphPad Prism 9.0 as mean ± SEM across 4 MMC treatment groups: control, Vit C, MMC, and MMC + Vit C. The pharmacological effects were shown as a side-by-side bar plot with each group with and without drug treatment. The *n* number in the single-cell calcium activity plots represents the number of cells accounted for. Three-to-five cells from each well of every treatment group were selected by the calcium-detection algorithm, with the highest fluctuation rate over 300 time-lapsed images, as described above. To establish the statistical differences between treatments groups, we performed a two-way ANOVA test, comparing each group’s mean with every other group’s mean by using Tukey’s statistical hypothesis with a family-wise threshold of α = 0.05 in 95% confidence interval. The statistical differences were represented as follows: * *p* < 0.05, ** *p* < 0.01, *** *p* < 0.001, and **** *p* < 0.0001.

## 3. Results

### 3.1. Macromolecular Crowding Treatment Enhances Deposition of ECM by Cultured Human-iPSC-Derived Astrocytes

Mature astrocytes secrete and deposit a wide variety of proteoglycans and extracellular matrix proteins which impact the growth of astrocytes themselves and contribute to the development of neural microenvironment and networks in vivo [33]. The neural ECM includes various interwoven meshworks of fibrillar proteins, including collagens, glycoproteins such as laminins, fibronectin, tenascins, and several classes of proteoglycans (heparan sulfate, chondroitin sulfate, dermatan sulfate, etc.) [34,36,37]. To investigate whether MMC has an impact on the extracellular matrix deposition by iAstros in vitro and, subsequently, on the activity of neural network in cultures, we designed an experiment, which is outlined in Figure 1. The iAstros were first seeded in wells of a 96-well microplate and treated for seven days with different combinations of MMC to induce ECM production, secretion, and deposition. The treatments tested were Ficoll 70/400 as a macromolecular crowder; L-ascorbic acid (Vit C) [38], which regulates collagen synthesis and secretion; and a combination of Ficoll 70/400 and Vit C (MMC + Vit C).

On day 8, we assessed ECM deposition, specifically collagen IV, fibronectin, and laminin (alpha1), using immunofluorescence microscopy. Representative images are shown in Figure 2a. We observed significant increases in the deposition of the three ECM proteins by the matured iAstros after 7 days of MMC treatment, as measured by the raw fluorescence intensity (RFU) of respective immunostainings, when compared against the control/no treatment group. The MMC treatment alone enhanced the collagen IV deposition 4-fold, (*p* < 0.0001), and the addition of Vit C to the MMC boosted the secretion of collagen IV to 7-fold (*p* < 0.001), whereas Vit C alone did not have any significant effect (Figure 2c). Similar observations were also made for fibronectin and laminin (Figure 2d,e). Fibronectin was increased by 1.5-fold (*p* = 0.002) with the MMC treatment and by 3.5-fold (*p* < 0.0001) with the MMC + Vit C treatment (Figure 2d). For laminin (alpha1), the deposition increased 1.5-fold (*p* < 0.0001) for MMC and 2.6-fold (*p* < 0.0001) for the MMC + Vit C group (Figure 2e). To assess whether any of these changes in the deposition of these ECM proteins were a consequence of iAstro proliferation or activation, we measured glial fibrillary acidic protein (GFAP) expression on astrocytes. After 7 days of culturing, only the iAstros treated with MMC + Vit C had an increase (30%, *p* = 0.0004) in GFAP signal compared to the untreated control group (Figure 2f).

We next investigated how the ECM deposited by the iAstros changed after the addition of neuronal cells. On day 8, iDopas were seeded on top of the iAstros–ECM bed and cultured for an additional 2 weeks. iDopas were strategically selected for the abundant in vivo works in the literature with pharmacological modulation studies which are possible to recapitulate in an in vitro model. On day 22, the iDopas + iAstros cocultures were assessed for collagen IV, fibronectin, and laminin (alpha1) deposition by immunostaining, as described above (Figure 2b). Collagen IV deposition was still higher in the MMC + Vit C group compared to the untreated group (2-fold, *p* < 0.0001), but there was a 20% (*p* = 0.0318) decrease in the amount of collagen IV in this treatment group compared to day 8 (Figure 2c). On day 22, fibronectin deposition was 2-fold (*p* < 0.0001) in MMC and 4-fold (*p* < 0.0001) higher in the MMC and MMC + Vit C treatments group, respectively, compared to the control group (Figure 2d). There was also a 20% (*p* = 0.0024) increase in fibronectin levels compared to day 8 for the MMC + Vit C treatment group (Figure 2d). On day 22, laminin (alpha1) deposition was 2.5-fold (*p* < 0.0001) increased with MMC + Vit C treatment compared to the control group, and there were no significant changes from the control for any of the other treatments (Figure 2e). After 2 weeks of neural coculture, the astrocytes under all four treatment groups appeared healthy by GFAP stanning, with small but significant increases in GFAP expression with the different treatments compared to the control: 10% (*p* = 0.026) in Vit C, and 16% for both MMC (*p* = 0.0014) and MMC + Vit C (*p* = 0.0009) group (Figure 2f). The qualitative comparison of the morphological of the ECM scaffold deposited before (day 8) and after (day 22) iDopas addition revealed that fibronectin and laminin (alpha1) appeared as mesh-like structures in the MMC + Vit C treatment group compared to the other treatments (Figure 2a,b, bottom row).

### 3.2. Macromolecular Crowding Treatment Induces Formation of Neuronal Extensions and Branching

The ECM has scaffolding functions to support cell attachment, growth, and migration [39]. However, there is also evidence that the ECM regulates many aspects of neural development, including the morphogenesis of neural tubes [40,41] and neocortex [42], and has a direct impact on the development of neural tissue [43]. To assess whether the MMC-induced ECM scaffold impacted any morphological features of the neuron–astrocyte cocultures, we used immunofluorescence microscopy and quantitative image analysis techniques. We first determined the number of dopaminergic neurons and astrocytes in our matured neural coculture from all four treatment groups. We then measured neurite length, longitudinal coverage of total neurites per well, total area coverage of astrocytes, and the average number of branching from the identified cell body as quantitative read-outs of neuronal morphology (Figure 3).

We assessed whether the MMC treatment affected the growth of astrocytes and dopaminergic neurons in terms of their cell counts. The iAstros and iDopas were marked with GFAP (Figure 3a, green fluorescence) and tyrosine hydrolase (TH) (Figure 3a, red fluorescence) antibody staining, respectively. The total cell nucleus was identified by Hoechst staining (Figure 3a, blue fluorescence). Our results revealed that there were no significant changes in the total or relative cell numbers of iDopas or iAstros in the neural coculture under any of the different treatment conditions (Figure 3b), except for the MMC-only treatment group, which had a small statistical increase (20%, *p* = 0.0117) in the iDopas count from its control counterpart; this could be biologically insignificant, though, considering the mean difference between the two groups. However, when the morphological features of the cells were analyzed, the data showed a statistically significant increase in the total area covered by iDopas under the MMC and MMC + Vit C treatments by 78.16% (*p* < 0.0001) and 79.44% (*p* < 0.0001), respectively (Figure 3c). We did not observe any significant alteration in the total area covered by the iAstros in neuron–astrocyte coculture under any treatment conditions (Figure 3c). To assess whether the increase in iDopas’ total area coverage reflected an increased formation of neurite extensions in the coculture, we used TH antibody staining for the fixed cells and green fluorescence signal from neural cultures transduced with a genetically encoded GCaMP6f biosensor for live cells to assess the live neurites’ length and numbers of branches. Representative images from all four treatment groups are shown in Figure 3a and Appendix A. An image-based algorithm was used to first identify the cell bodies from the GCaMP6f green fluorescence signal and Hoechst staining for the fixed cells (Appendix A), and then the neurite length and branching (Appendix A multicolor) were located and measured with image-processing software. The combined quantitative analysis shown in Figure 3d indicated that the average longest neurites found within a well (max neurites length) increased 2-fold (*p* = 0.0012) under MMC alone and 4-fold (*p* < 0.0001) under the MMC + Vit C treatment. The total coverage of neurites (total neurites length) increased by 3-fold (*p* = 0.0474) in the MMC group and 10-fold (*p* = 0.0124) in the MMC + Vit C treatment group. Our MMC + Vit C treatment also strengthened the neuronal network by increasing the average number of neurites branching by 3-fold (*p* = 0.0055). We did not see any statistically significant change in neurite branching for the MMC-alone or Vit C–alone treatment groups.

### 3.3. Macromolecular Crowding Treatment Induces Synaptogenesis and Synaptic Plasticity

The observation of neurites’ outgrowth led us to investigate the effect of MMC-induced ECM secretion on the synaptic properties of the neural networks. Presynaptic activity was assessed with a synaptic dye which fluoresces green only when embedded into the newly formed transmitter vesicles at the presynaptic ends of the neurons. This dye has a lipophilic tail at one end and is highly hydrophilic at the other end. In the aqueous phase, the dye is non-fluorescent, but during endocytosis following neuronal action potentials, it is trapped in the lipid membrane of the transmitter vesicles by its lipophilic tails and becomes intensely fluorescent. The fluorescent signal is proportional to the number of newly formed synaptic vesicles. We used this dye to trace the activity at the presynaptic end of our dopaminergic neurons for the four different treatment conditions. The representative confocal images shown in Figure 4a demonstrate an increase in live transmitter vesicle density with MMC treatment. The quantitative analysis is shown in Figure 4b. Presynaptic fluorescence intensity (normalized by the background fluorescence) increased by 1.7-fold for the MMC treatment group (*p* = 0.0017) and 2.4-fold (*p* < 0.0001) for the MMC + Vit C treatment group. We further looked at the formation of postsynaptic terminals by immunostaining, using PSD95 marker. Figure 4c shows the representative confocal images of PSD95 (magenta) that were co-stained with either astrocytes marker GFAP (Figure 4c, top row, green) or with neurites projections marker MAP2 (Figure 4c, bottom row, yellow). The combined quantitative analysis from both of these double-staining procedures (Figure 4d) show an increase in PSD95 density (dF/F_0_) by 20% (*p* = 0.0113) with MMC treatment and by 80% (*p* < 0.0001) with the MMC + Vit C treatment, compared to the control group. The postsynaptic density on astrocytes increased by 2.25-fold (*p* = 0.0021) and 2.78-fold (*p* < 0.0001) (Figure 4e) and on neurons by 1.58-fold (*p* < 0.0001) and 1.95-fold (*p* < 0.0001) (Figure 4f) after MMC and MMC + Vit C treatment, respectively, from their respective control group. Though the postsynaptic density enhanced more on astrocytes than neurons after the MMC treatment from its untreated counterparts, the neurites’ outgrowth boosted the overall postsynaptic connections on neurons substantially compared to astrocytes. We observed 27.34% and 33.71% PSD95 co-localized with GFAP, while 74.27% and 91.53% were with MAP2 in the MMC and MMC + Vit C groups (Figure 4e,f). Collectively, these data indicate that induced ECM secretion by astrocytes upon treatment with a MMC together with L-ascorbic acid accelerated the development of a mature complex neuronal network with improved synaptic connections.

### 3.4. Macromolecular Crowding Treatment Upregulates Neural Calcium Dynamics

To assess whether the accelerated development of a mature neuronal network with improved synaptic connections seen in MMC + Vit C treatment translated into functional modulation of neural activity, we measured the calcium dynamics of genetically encoded GCaMP6f biosensor by using the single-cell fluorescence imaging method in our coculture. Representative confocal images of the GCaMP6f activity from all four treatment groups are shown in Figure 5a. The oscillations of calcium waves (dF/F_0_) from four randomly chosen example traces from each experimental group are shown in Figure 5b. Figure 5c is the heat map representation of the collective ROIs from the respective treatment groups over time. Each row of the heat map represents a single cell’s (ROI) activity over time. The blue-to-yellow color scale in Figure 5c indicates the increases in amplitude as dF/F_0_. The peak frequencies (peak per s), mean peak amplitudes (dF/F_0_), and mean peak width (ms) were calculated for each ROI, with a minimum of 3 ROIs per well, 3 wells per group, and 3 biological replicates, and plotted as bar graphs (Figure 5e). The data show that there were significant increases in calcium peak frequency by 30% (*p* = 0.01), 50% (*p* = 0.0001) and 70% (*p* < 0.0001) in the Vit C, MMC, and MMC + Vit C treatments, respectively. We did not observe statistically significant changes in peak amplitude and peak width between the different treatment groups because of scattered peak property values. We also assessed the synchronicity of single cellular calcium dynamics within the network by plotting a correlation heat map of ROIs vs. ROIs in Figure 5d, and we calculated a correlation score within group, shown as bar plots in Figure 5f, to statistically compare the synchronicity between groups. The color gradient from dark to light brown indicates the synchronicity from −1 to +1. Synchronicity in spontaneous calcium oscillations per cell is a measure of the overall network’s functional dynamics of neural cultures. The correlation heat maps display an overlap in peak frequency between single neuron’s (ROI) calcium oscillations. We observed 2.71% and 7.61% lower synchronicity in the calcium oscillations with the MMC (*p* = 0.0478) and MMC + Vit C (*p* < 0.0001) treatments, respectively, compared to the untreated group.

### 3.5. Macromolecular Crowding Treatment Promotes Pharmacological Responses That Emulate Those Seen In Vivo

We further explored whether the differences in neural activity seen for the MMC-treated groups translated into more relevant pharmacological responses that emulate in vivo data. We chose three drugs targeting the D2 receptor expected to be found on iDopas neurons: apomorphine, a D2-receptor agonist; metoclopramide, a D2-receptor antagonist; and morphine, a µ-opioid receptor agonist. On day 22, the single-cell calcium fluorescence was first acquired (Figure 6a–c, empty columns) for each neural coculture, and then compounds (individually) were added to each well and incubated for 5 min, followed by repeat acquisition of single-cell calcium fluorescence (Figure 6a–c, filled columns). The top row of Figure 6–c showed bar plots for the calcium peak frequencies before (empty columns) and after (filled columns) the addition of apomorphine, metoclopramide, and morphine respectively, for the four MMC treatment groups. The detailed calcium peak traces and analysis of each individual drug treatment are presented in Supplementary Appendix A for apomorphine, metoclopramide, and morphine, respectively.

The graphs at the bottom row represent the % change in peak frequency $\left[\left(F_{drug}-F_{no\;drug}\right)/F_{no\;drug}\right]$ for each compound treatment, in each treatment group. For apomorphine, we observed a ~2-fold increase in calcium peak frequency over the no-drug control for the no-treatment group (*p* = 0.0002) and Vit C groups (*p* = 0.0215), but a decrease of 40% (*p* = 0.0491) in the MMC + Vit C group (Figure 6a), and there was no statistical change in the MMC-treatment group. As described above, the ROIs vs. ROIs correlation heat map (Appendix A), before apomorphine exposure, indicated that the neural network calcium synchronicity decreases with MMC treatment, a phenomenon that was reversed by treatment with apomorphine. We observed low synchronous activity in the control group after apomorphine exposure, and the synchronicity gradually increases in the Vit C to the MMC to the MMC + Vit C group. Exposure to metoclopramide (Figure 6b) increased the peak frequency by 60% (*p* = 0.0096) in the control group and 30% (*p* = 0.0253) in the MMC + Vit C group, but there were no significant changes observed for the Vit C and MMC groups. There were also no notable effects observed in the synchronicity of the calcium oscillation before and after treatment with metoclopramide (Appendix A). Similar effects on calcium peak frequency were observed for morphine treatment (Figure 6c). There was an increase of 100% (*p* < 0.0001) for the control group, 24.6% (*p* = 0.0465) for the MMC group, and 40% (*p* = 0.0001) for the MMC + Vit C treatment group. There was no statistically significant change observed for the Vit C treatment group. As seen with apomorphine (D2-receptor agonist), morphine also increased synchronicity in calcium waves in the MMC + Vit C treatment group compared to its control counterpart (Appendix A).

## 4. Discussion

The advent of iPSC-derived cells has enabled the ability to develop neural cellular models with patient-derived cells. In most cases, cells are grown on plastic surfaces coated with exogenously added biopolymers that mimic native ECM to enhance the growth, differentiation, and function of the neural cultures. However, these coating do not match the complexity of neural ECM, including various interwoven meshwork of fibrillar proteins such as collagens, glycoproteins such as laminins, fibronectin, tenascins, and several classes of proteoglycans (heparan sulfate, chondroitin sulfate, dermatan sulfate, etc.) [34,35,37]. Therefore, to create more physiologically relevant cellular cultures, it would be more desirable to develop cellular models which synthesize, secrete, and deposit their own native similar to ECM. This can be achieved by exposing adherent cell cultures to the tissue engineering technique of macromolecular crowding [23]. This technique has been applied successfully in mesenchymal and epithelial cells and tissue equivalents [23,44] but has been hitherto not been explored in neuronal and neuroglial cells. Here we show that the simultaneous addition of Ficoll 70/400, a known macromolecular crowding agent, and L-ascorbic acid (vitamin C), a crucial cofactor for collagen secretion, significantly enhances deposition of brain-relevant ECM components by astrocyte monolayer cultures. In this initial experimental design, astrocytes were chosen as targets for the MMC because they are critical for ECM formation in the brain [45]. Mature astrocytes secrete and deposit a wide variety of proteoglycans and extracellular matrix proteins which impact the growth of astrocytes themselves and contribute to the development of the neural microenvironment and networks [33]. However, most of what is known on astrocyte–ECM production is derived from pathological processes such as glial scar formation and spinal cord injury [46]. The physiological ECM-building capabilities of human astrocyte production in vitro have been under researched. We therefore designed our experiments by first pretreating astrocytes with MMC and Vit C to create an ECM bed on which neurons could attach and more effectively establish active functional networks. Ascorbate plays a vital role in brain function, and it is commonly added as a supplement in neuronal culture medium [31,47,48]. It has also been used in matrix biology to induce collagen secretion by fibroblasts [49], but we found only one report of ascorbic acid being used to induce matrix secretion by astrocytes [38]. Our unpublished preliminary experiments with several reported macromolecular crowders, including Dextran sulfate (DxS 500 kDa), Poly-sodium 4-styrene sulfonate (PSS 200 kDa), and Polyvinylpyrrolidone (PVP 360 kDa), for these astrocyte cultures showed that the largest increase in secretion of collagen IV, fibronectin, and laminin (alpha1) was obtained with Ficoll 70/400. These proteins were chosen because they have been shown to be relevant for formation of neuronal extensions and networks [6,43] and are detectable by immunostaining. Further work using alternative technologies such as MALDI mass spectrometry will be conducted in the future to explore the deposition of additional ECM proteins and proteoglycans. Our results show denser ECM proteins’ deposition by the human astrocytes in vitro after MMC treatment. Published work has shown that the production of ECM and activation of astrocytes accompanies CNS remodeling during development and following injury [50,51]. In our experiments, induction of ECM secretion with MMC and ascorbic acid induced small but significant changes in expression of GFAP on astrocytes. Moreover, when culturing the neurons on top of the MMC-treated astrocytes, the filament-like ECM mesh remarkably aligns itself with mature neurites’ outgrowth, creating a morphological microenvironment similar to neural cocultures in 3D gel-based scaffold (Kundu et al., 2022, manuscript under review) [52,53].

Work in *C. elegans* suggests that ECM composition surrounding synaptic terminals modulates neuronal morphology and synaptic integrity [54]. In the present study, we established that MMC-enhanced deposition of collagen IV, fibronectin, and laminin in cultured astrocytes indeed promoted dopaminergic neuronal growth, neurites’ branching pattern, and the strengthening of synaptic connections, as evidenced by both presynaptic activity and postsynaptic density. Increased synaptic density for astrocytes might contribute to regulating synaptic plasticity and calcium signaling cascades [55,56]. Previous report shows that laminin at the synaptic cleft modulates synaptic plasticity and controls calcium activity [57]. The effects of the ECM on neural activity could be caused by enhanced synapsis and network formation, as was observed, or by modulation of ions and other factors that affect action potentials. ECM is the largest source of free calcium ions in the living multicellular organism, with 10^4^ times higher concentration than cytosol [58], and there are also reports that show that several ECM components can regulate cytosolic calcium levels via mechanotransduction [59]. For example, fibronectin induces cytosolic calcium spikes [60,61]. Laminin also has calcium binding sites, and a recent study shows direct interaction of laminin with voltage-gated calcium channel in neuro-muscular junctions [57,62]. Lastly, studies using human mesenchymal stem cell culture shows that calcium spikes increase with the increasing stiffness of the collagen matrix [63]. These three components of ECM were elevated in our system after MMC and MMC + Vit C treatment. We therefore sought to know whether the observed changes in ECM and in network plasticity translated into modulation of neural network functionality. Our data show that augmented PNNs-like ECM deposition under MMC + Vit C treatments increases the frequency of calcium oscillations, while decreasing the overall synchronicity of the neural network in vitro. There are studies showing that the PNN structure regulates brain activity in vivo by modulating synaptic plasticity [64]. Irregular synchrony and oscillation in dopaminergic network under stress conditions observed in schizophrenia patients [65], as well as in rodent model [66], lead us to cortical–hippocampal–amygdala informational pathway [67], where degradation of PNNs modulates fast, spiking the GABAergic interneuronal firing rate by generating gamma waves underlying synchronous network activity, suppressing cognitive function and emotional processing [68,69]. However, the direct evidence of PNNs influencing dopaminergic neuronal oscillation in vivo has yet to be found. We also observed low network synchronicity in calcium oscillation in 3D-hydrgel-based neural cultures, using the same iAstros and iDopas (Kundu et al. 2022, manuscript under review), which again would suggest that, by using MMC to enhance ECM deposition, we are creating a more three-dimensional (3D)-like microenvironment that affects the intrinsic properties of network connectivity and, hence, enhances the complexity of neuronal network toward in vivo relevance.

One of the goals of developing physiologically relevant cellular models is their use as predictive assays for drug development. Therefore, it is important to establish whether pharmacological responses in cell cultures building their own native ECM are more similar to those seen in vivo than those obtained in standard 2D cultures. D2 receptors are abundant on the pre- and postsynaptic terminals of dopaminergic neurons. They respond to dopamine released by their own nerve ending, thereby negatively modulating the dopaminergic neuronal activity; thus, they function as autocrine inhibitory receptors [70]. In vivo studies show that the D2-receptor agonist apomorphine inhibits local field potentials of dopaminergic neurons in rat substantia nigra [71,72]. In agreement with these results, our study showed that apomorphine reduced calcium oscillations’ frequency in the MMC + Vit C treated conditions. This contrasts with the effects seen in astrocyte–neuron cocultures without prior MMC + Vit C treatment which showed a positive calcium modulation upon stimulation with apomorphine. Therefore, our data suggest that the MMC-enhanced astrocytic ECM microenvironment can make certain pharmacological responses more similar to in vivo. Interestingly, we have observed a similar pharmacological modulation of calcium activity in iDopas and iAstros coculture system in a 3D fibrinogen gel matrix (Kundu et al. 2022, Manuscript under review). Correspondingly, metoclopramide had the opposite effects to apomorphine and increased calcium oscillations’ frequency in the MMC + Vit C treatment group. Several studies in rodent models have shown that morphine activates dopaminergic neuronal excitation [73]. Similarly, our data showed that morphine induced an increase in calcium oscillations, regardless of the treatment conditions. It is also interesting to point out that the magnitude of modulation under pharmacology intervention in the no-treatment/control condition was, in general, larger than in the presence of enhanced ECM deposition, suggesting that the induction of ECM might limit diffusion of the drug to the site of action [74,75]. The molecular and cellular changes that the iDopas undergo in a pseudo 3D environment created by the MMC + Vit C treatment compared to a 2D environment would explain the in vivo–like pharmacological responses, but these changes, perhaps dopamine-receptor levels, trafficking, or dopamine levels in the synapses, are still being investigated. Overall, our data suggest that the pharmacology of agonists of the dopamine receptor is more native-like in the conditions induced by addition of MMC and vitamin C to cultured iAstros.

Taken together, utilizing macromolecular crowder in combination with an enhancer of collagen secretion, such as L-ascorbic acid, we showed that human iPSC-derived astrocytes can be induced to deposit a more brain-relevant extracellular matrix and that the resultant matrix favors the development of iPSC-derived neurons to form active synaptic connections neuronally and enhance neuronal activity, thus altering the intrinsic physiological property of the neuronal network in terms of synchronicity. This study provides a glimpse of in vivo–relevant neuronal modulation in vitro when intrinsic matrix formation is facilitated by MMC. This approach is demonstrated in a 96-well plate platform that is compatible with a high-content-drug-screening format and offers applications with physiological in vitro models that more closely mimic properties of the brain in vivo.

## Figures and Tables

**Figure 1 cells-11-02131-f001:**
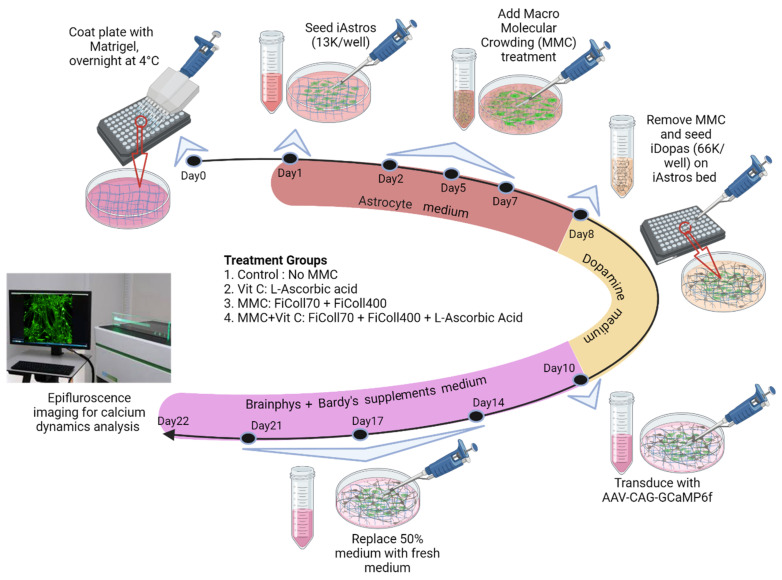
Schematic flow diagram of human iPSC derived human dopaminergic (iDopas) and astrocyte (iAstros) coculture treated with various combination of macromolecular crowding. The flow diagram shows the timeline for the macromolecular crowding treatment on the iDopas + iAstros coculture system. The PerkinElmer Ultra clear-bottom 96-well plates were first coated with Matrigel at 4 °C overnight, followed by seeding of iAstros (1.3 × 10^4^/well) the next day, and then they were incubated at 5% CO_2_, at 37 °C. Four combinations of macromolecular crowding testament were used: (1) control, cells did not receive any crowding treatment; (2) Vit C, cells received 100 µM of vitamin C; (3) MMC, cells received cocktail of 37.5 mg/mL Ficoll70 and 25 mg/mL Ficoll400; and (4) MMC + Vit C, cells received same cocktail of Ficoll70 and Ficoll400, along with 100 µM of vitamin C. With every 100% of astrocyte medium changed, the cells received one of the four treatments (*n* = 3 wells/group; plates in duplicate). After 24 h of cell seeding (marked as day 1), cells were treated with MMC at day 2, day 5, and day 7. At day 8, one culture plate was fixed with 4% PFA for immunohistochemistry study, and a duplicate plate was carried forward with the seeding of iDopas neurons (6.6 × 10^4^/well) on top of the treated astrocyte–ECM bed. The neurons were seeded after passing through dead cell removal process with a minimum viability of 85%. Caution was taken to replace (100%) astrocyte medium containing MMC with iDopas neuronal medium. After 2 days of iDopas seeding (at day 10), the iDopas neuronal medium was 100% replaced with Brainphys medium + Bardy supplement for better neuronal growth. The coculture was then maintained for 14 more days at 37 °C, with 5% CO_2_. At day 22, all the functional assays were performed (*n* = 3 technical replicates × 3 biological replicates/treatment group).

**Figure 2 cells-11-02131-f002:**
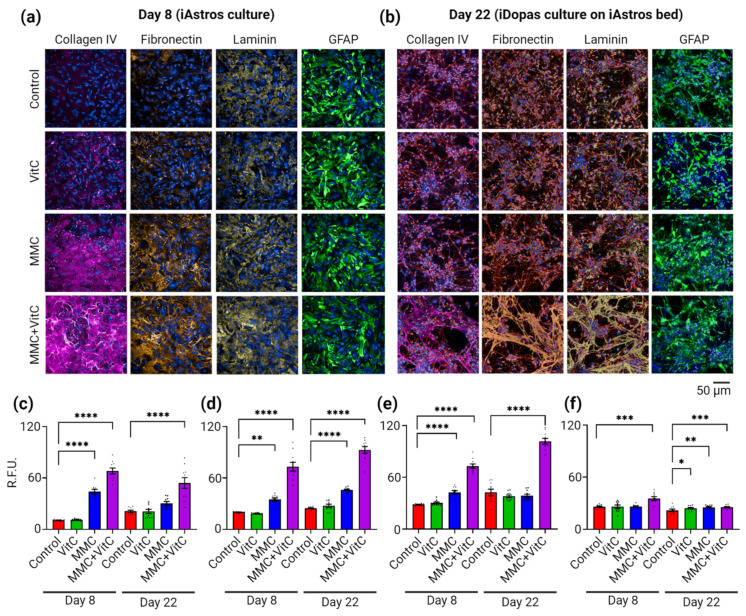
Macromolecular crowding enhances extracellular matrices’ deposition by iAstros. Comparative analysis of extracellular matrix deposition was performed by immunostaining and fluorescence microscopy imaging. All four groups (control, Vit C, MMC, and MMC + Vit C) were stained systematically with anti-human collagen IV, fibronectin, laminin (alpha1), and GFAP antibodies. (**a**,**b**) Representative images of extracellular matrices’ formation by iAstros monoculture at day 8, and in the iDopas and iAstros neural cocultures at day 22, respectively, with and without MMC treatments. (**a**) Macromolecular crowding of Ficoll70/400 (third row from top) induced deposition of collagen IV (magenta), fibronectin (orange), and laminin-alpha1 (yellow) at day 8, in comparison to the control group, and it was further enhanced with the addition of Vit C (bottom row). Vit C alone could not make any significant difference. The iAstros (GFAP staining) appeared healthy and matured in all four groups. (**b**) The iDopas were seeded on top of the treated astrocytes–ECM bed, without disturbing the deposited extracellular matrices, at day 8 and cocultured for another 14 days. At day 22, the matrices were stained with the aforementioned antibodies. The MMC + Vit C group was, again, the most supportive of ECM secretion (bottom row). A high density of web-like patterns was apparent from all three stainings, collagen IV (magenta), fibronectin (orange), and laminin (yellow), which may assist in the formation of neuronal network. The iAstros phenotypes were checked again with GFAP staining (green). This web-like pattern was not profound on any other of its counterparts (top three rows). Nuclei were stained with Hoechst in blue. Scale bar: 50 µm. (**c**–**e**) Bar graphs for the quantifications of deposited extracellular matrices density in terms of the mean raw fluorescence intensity (RFU) for collagen IV, fibronectin, and laminin, for each experimental group. (**f**) The RFU of GFAP staining shows the healthy growth of iAstros after day 8 and day 22 in all four groups. The bar plots (*n* = 3 wells/group; 3 biological replicates) showed that all the extracellular depositions significantly enhanced after MMC + Vit C treatment. The bar plots are mean ± SEM. Only statistically significant differences calculated by two-way ANOVA were marked with * *p* < 0.05, ** *p* < 0.01, *** *p* < 0.001, and **** *p* < 0.0001.

**Figure 3 cells-11-02131-f003:**
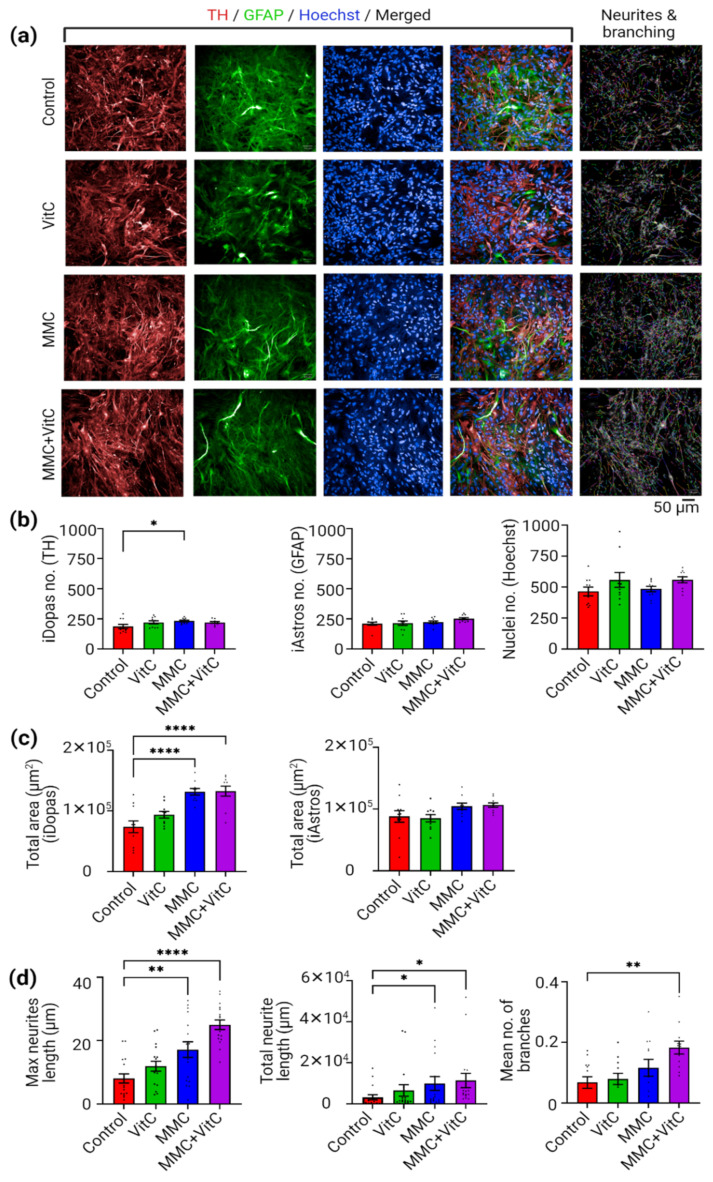
Macromolecular crowding alters the phenotype of neural network. Image-based analysis was used to determine morphological differences in terms of the neurites’ length and number of neurites branching between all four macromolecular crowding groups (control, Vit C, MMC, and MMC + Vit C). (**a**) The iDopas + iAstros cocultures from all four MMC treatment groups imaged under 20X water immersion confocal microscope after immunostaining with cell-specific primary antibodies. Anti-TH anybody (red) and anti-GFAP antibody (green) used to marked iDopas and iAstros cells, respectively, along with Hoechst 33342 (blue) for localization of the nuclei. Image analysis of neurites extension and branching pattern (multicolored) by Harmony v5 shows that MMC + Vit C group formed a complex network with long and branched iDopas projections, compared to other groups (last column). (**b**) The “find cell” algorithm was applied individually to each staining, TH, GFAP, and Hoechst 33342, and shows no significant changes in the number of cells per well in each treatment group. (**c**) Quantitative analysis of total area coverage by the iDopas (TH staining) and iAstros (GFAP staining) indicates that, with the MMC and MMC + Vit C treatments, the neuronal covered area increases without any changes in astrocyte-covered area. (**d**) The “find cell” algorithmic script for image analysis in Columbus software (PerkinElmer), providing an ROI of ≥20 µm^2^, was used to locate the neuronal cell body, and the CSIRO Neurites Analysis 2 method was applied to mark the length, along with segment/branches of neurites that originated from those preselected cell bodies. Quantitative estimations of various parameters, including maximum neurite length (µm), total neurites length (µm) per well, and mean number of branches, were compared among all four MMC treatment groups. The enhancement of extracellular matrices’ deposition by iAstros under MMC + Vit C treatment supported the formation of denser iDopas’ neuronal network with significantly increased neurites’ length, total neurites coverage, and increased number of branches. The bar plots are mean ± SEM (*n* = 3 wells/group: 3 biological replicates). Scale bar: 50 µm. Only statistically significant differences calculated by two-way ANOVA marked with * *p* < 0.05, ** *p* < 0.01, and **** *p* < 0.0001.

**Figure 4 cells-11-02131-f004:**
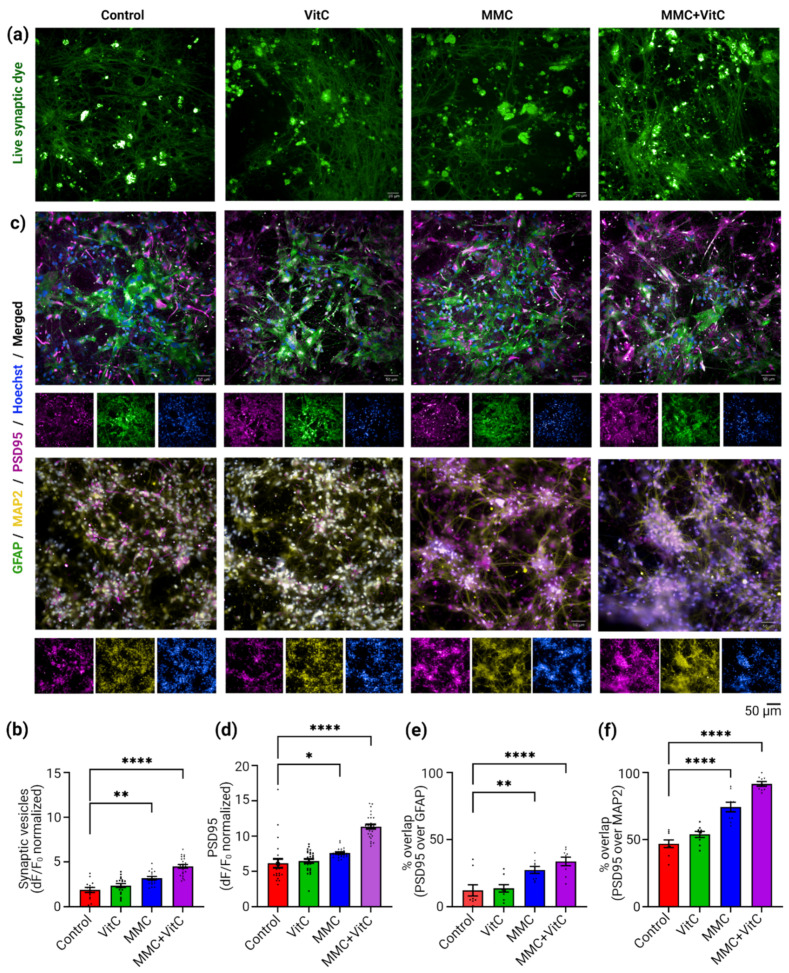
Macromolecular crowding enhances neuronal synaptic plasticity. Macromolecular crowding with Vit C strengthened the neural network by enhancing the formation of presynaptic vesicles and increasing postsynaptic connections. Quantitative assessment of neuronal synaptic plasticity among all 4 MMC treatment groups performed in two ways. (**a**) Live cell imaging of newly formed active synaptic vesicles inside of presynaptic end of the neurons marked with BioTracker 510 Green C1 Synaptic Dye. After 10 min of incubation with the dye at room temperature, live cell epifluorescence images were captured under 10X water objective, with FITC (λ_Ex_ = 490 nm; λ_Em_ = 530 nm) filter, from all four groups (control, Vit C, MMC, and MMC + Vit C) represented here. The newly formed neurotransmitter vesicles at the cells synaptic terminals appeared as green-fluorescent puncta as the lipophilic tail of the dye was embedded into the vesicle’s membrane. (**b**) The mean raw fluorescence intensity (RFU) normalized by their respective background fluorescence from each group (3 fields of view/well; *n* = 3 wells/group from 3 biological replicates) quantified with by ImageJ analysis software and plotted as a bar graph. (**c**) The postsynaptic density of neural network was estimated by immunostaining method with Goat anti mouse PSD95 antibody (magenta) for all four groups. PSD95, the post-synaptic markers were co-stained with either GFAP (green) or MAP2 (yellow) antibodies to assess the presence of synaptic density on both iAstros and iDopas, respectively. Representative confocal images (10X water objective) of day 22 neural co-cultures from all four groups were presented here. (**d**) Mean fluorescent intensity of PSD95 (magenta) normalized by the background florescence from both sets of co-staining (3 field of views/well; 6 wells/group) were quantified by using ImageJ software and are displayed as a bar plot. (**e**,**f**) Bar plots for the percent area overlap of PSD95 with GFAP and MAP2, respectively. Scale bar: 50 µm. The bar plots are mean ± SEM. Only statistically significant differences were calculated by two-way ANOVA marked with * *p* < 0.05, ** *p* < 0.01, and **** *p* < 0.0001.

**Figure 5 cells-11-02131-f005:**
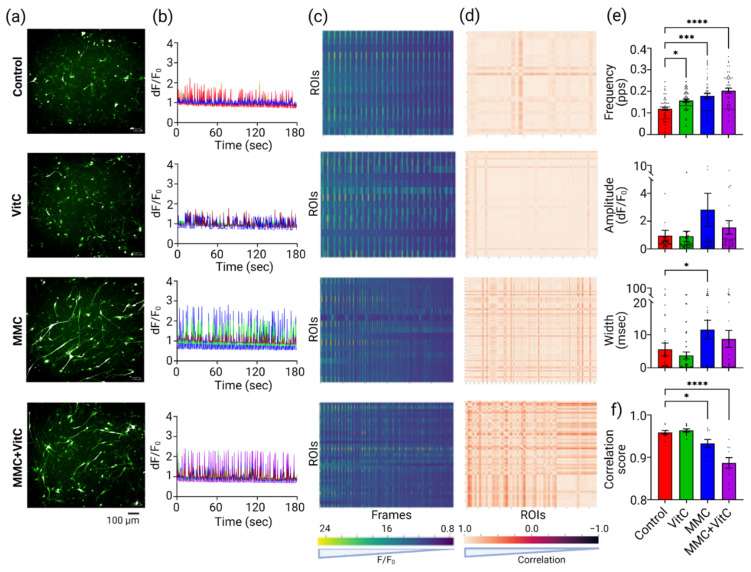
MMC-induced ECM upregulates calcium flux in neural cocultures. We measured the functionality of iDopas + iAstros coculture network by imaged-based single-cell calcium flux, using AAV-CAG-GCaMP6f transfection. (**a**) Representative epifluorescence (10X water objective) images of iDopas + iAstros coculture transfected with AAV6-CAG-GCaMP6f. The expression of GCaMP6f in the neuronal network is shown in green FITC (λ_Ex_ = 490 nm; λ_Em_ = 530 nm). (**b**) Representative traces of single-cell neuronal calcium dynamics over time for each treatment groups are shown. A series of 300 time-lapsed images were captured over a period of 180 s, with 0.6 s intervals, and single-cell calcium flux (dF/F_0_) was estimated by “LC-Pro” plug-in in “ImageJ”, as described in the Method section. (**c**) The frequencies of the calcium peaks from 50 individual neurons (Regions of Interest) over time (frames) were also plotted as a heat map, with increasing amplitude (dF/F_0_) from 0.8 to 2.4 (blue to yellow). (**d**) Correlation plots are shown as heat maps for the 50 cells/ROIs against each other. Color gradient from darker brown to lighter brown shows in color scale as negative 1 to positive 1 correlation. Correlation score −1 would indicate corresponding neurons firing completely out of phase, whereas correlation score +1 would indicate neurons firing in synchrony. (**e**) Quantified calcium peak properties, peak frequency (pps), peak amplitude (dF/F_0_), and peak width (ms) from 9 wells (*n* = 3 wells/group, 3 biological replicates) are represented as bar plots. (**f**) Bar graph of correlation scores. Scale bar: 50 µm; The bar plots are mean ± SEM. Only statistically significant differences calculated by two-way ANOVA marked with * *p* < 0.05, *** *p* < 0.001, and **** *p* < 0.0001.

**Figure 6 cells-11-02131-f006:**
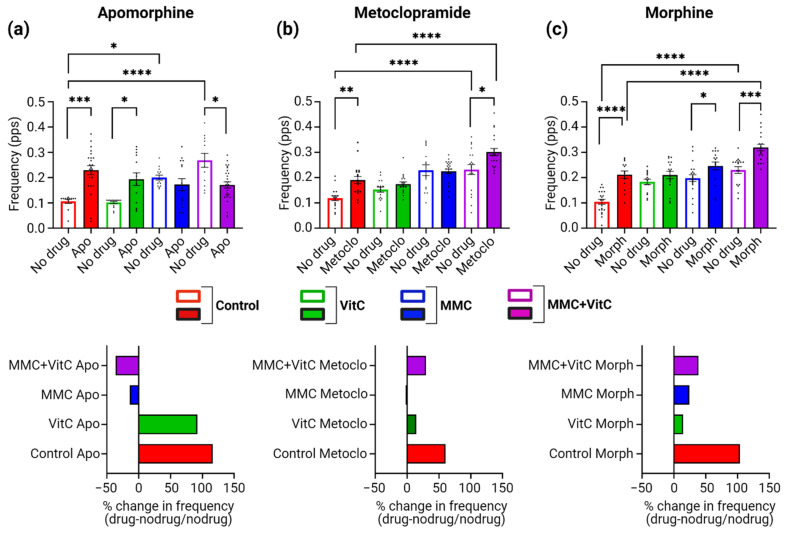
Pharmacological perturbation of neural calcium dynamics on macromolecular-crowding-treated neural cocultures. Pharmacological perturbation of single-cell fluorescence calcium dynamics demonstrated with (**a**) D2-receptor agonist (apomorphine), (**b**) D2-receptor antagonist (metoclopramide), and (**c**) µ-opioid receptor agonist (morphine) on all four MMC treatment group of iDopas + iAstros coculture. Top row represents the calcium peak frequency (pps) bar plots comparing before and after pharmacological interventions in all 4 MMC treatment groups. Bottom row shows the percent change in peak frequency after individual pharmacological interventions from the corresponding no drug basal level. The bar plots are mean ± SEM from 9 wells (*n* = 3 wells/group, 3 biological replicates). Only statistically significant differences calculated by two-way ANOVA marked with * *p* < 0.05, ** *p* < 0.01, *** *p* < 0.001, and **** *p* < 0.0001.

**Table 1 cells-11-02131-t001:** Primary and secondary antibodies and dilutions.

Antibody Target (Clone)	Primary/Secondary	Host and Clonality	Manufacturer and Catalog No.	Dilution
Collagen IV	Primary	Mouse monoclonal	Dako, M0785	1:200
Fibronectin	Primary	Rabbit polyclonal	Abcam, ab2413	1:200
GFAP	Primary	Rabbit polyclonal	Dako, Z0334	1:1000
Laminin (alpha1)	Primary	Rabbit polyclonal	Dako, Z0097	1:200
Microtubule-associated protein2 (MAP2)	Primary	Mouse monoclonal	Sigma, M4403	1:200
Tyrosine hydroxylase (TH)	Primary	Rabbit polyclonal	EMD Millipore, AB152	1:1000
Goat anti-mouse 555	Secondary		ThermoFisher, A28180	1:500
Goat anti-rabbit 647	Secondary		ThermoFisher, A27040	1:500
Goat anti-chicken 488	Secondary		Invitrogen, A32931	1:500

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
