# Peer review of "Enhancement of Neuroglial Extracellular Matrix Formation and Physiological Activity of Dopaminergic Neural Cocultures by Macromolecular Crowding"

_cells, 2022, doi:10.3390/cells11142131_

Round 1

Reviewer 1 Report

Manuscript ID: cells-1747523

Comments to the author:

The manuscript by N. Vo. et al. highlights the importance of in vivo-like ECM traits in in vitro models of neuroglial interactions. Leveraging the macromolecular crowding (MMC) technique, they provide evidence that a) astrocytes produce ECM-relevant proteins in the MMC conditions, b) neurons co-cultured with MMC-treated astrocytes form more complex networks supported by neuronal extensions and branching and show increased synaptogenesis, plasticity, and calcium activity and c) the MMC-treated astrocyte-neuronal cell cultures emulate physiological responses to common compounds targeting dopamine receptors in the brain. Overall, the manuscript is well written, emphasizing details and solid methodology; yet several aspects should be improved before publication. I appreciate the detailed description of image analysis methods that were followed.

Detailed comments

Introduction

1.     MMC is covered; however, this part should include some basic literature regarding physiological responses of neurons to compounds (both agonists and antagonists). In addition, in vivo and in vitro studies should be included.

2.     Some sentences lack references. Please review and add supporting references; for example, ECM proteins supporting neuronal growth and extensions in 3D engineered microenvironments should be described and referenced.

3.     Describe the importance of the cellular sources and their benefits. 

4.     The importance of the mesh-like structures should be further highlighted

Methods

5.     The statistical analysis section should include accurate definitions of biological and technical replicates.

Results

6.     Line 297. Explain bio and tech replicates. What are the differences between the two plates? What is a bio replicate in the experiments? These should be clearly explained in the methods and in the figures.

7.     Lines 308-310. The authors should support the relevance of the proteins with literature and explain which interactions/processes they are modelling. They should support their data with both animal and in vitro studies.

8.     Lines 321-323. Were the cell numbers quantified for the GFAP+ iAstros?

9.     Remove unnecessary parts of the graphs in all figures. Indicate only significant results to improve figure appearance and readability of the manuscript (remove ns and state it in the legend).

10.  Figure 2. Morphologically astrocytes look different at d22. Can the authors provide gene expression and/or branching/processes analysis for the iAstros at d22

11.  Compound testing experiments: Indicate expected/non-expected results and why. Why D2R and what is expected post MMC treatment? The introduction should include some basic information regarding perturbations and how calcium dynamics reflect neuronal functionality. The current text lacks the motivation for the follow up experiments in some parts.

Discussion

12.  Lines 601-603: Missing references

13.  Lines 618-619: If there are any studies for VitC in neurobiology context, please cite them

14.  The authors often have a manuscript under review (Kundu et al 2022) to support their claims. I understand the need to incorporate recent findings, and it is acceptable, but are there any additional studies supporting their statements in the respective fields?

15.  The decrease in synchronicity and the physiological relevance post MMC should be discussed further. Why does this reflect better the in vivo state? (plus citations)

16.  The morphine results are not discussed

Minor concerns

There are typos throughout the manuscript; the text should be carefully revised and corrected.

Reviewer 2 Report

In the manuscript “Enhancement of of Neuroglial Extracellular Matrix Formation and Physiological Activity of Dopaminergic Neural Cocultures by Macromolecular Crowding” Vo et al. study the role of macromolecular crowding with vitamin C to recapitulate ECM formation and its natural physiology in co-cultures of astrocytes and dopaminergic neurons.

The  description of the techniques used are carefully described, the results are presented in text and in illustrative figures. Both introduction and discussion help to understand the background and the significance of the results obtained. However, there are some issues that must be resolved before considering this article for publication.

Major issues:

In lines 321-323 the authors stated that “None of the MMC treatments induced large changes in astrocyte numbers and maturity, after 7 days, as shown by GFAP staining (Figure 2f), compared to the  control group.”. Although literally true, it omits the fact that there is a difference in the GFAP signal (rfu). Also, the actual number of astrocytes is given only later in Fig.3, and maturity of the astrocytes is not given, and it might be difficult to estimate based on the fluorescence intensity of GFAP staining alone. Later, it is said in lines 366-368 that “After 2 weeks of  neural co-culture, the astrocytes under all four treatment groups appeared matured by GFAP stanning, without significant change in r.f.u. (Figure 2f)”. However, figure 2f shows significant differences for day 22. Authors could perform additional experiments addressing this issue, including some to estimate the effect of small increases in the number of astrocytes in the results obtained, that further validate their previous statement, and correct their statements when necessary.

Later the authors also stated (in lines 416- 418) that “Our results revealed that there were no significant changes in the total or relative cell numbers of iDopas or iAstros the neural co-culture under any of the macromolecular crowding treatment conditions (Figure 3b).” Figure 3b seems to indicate a difference in the number of TH cells between control and MMC. Please, explain.

Minor comments:

In the section “3.3. Macromolecular Crowding Treatment Induces Synaptogenesis and Synaptic Plasticity” the authors analyze the effects of the different treatments on postsynaptic density, and found that “Though the post-synaptic density enhanced more on astrocytes than neurons after the MMC treatment from its untreated counterparts” How do you interpret this, and what is its relationship with the pharmacological responses observed later in the manuscript.

The authors indicated in the discussion (lines 616-618) that “VitC has long been used in matrix biology to maximize collagen secretion by fibroblasts, but for the exception of Cheng et al (2020), which used it to produce astrocyte matrix, supplementation of neural cultures with VitC appears only rarely in neurobiology literature.” I would encourage the authors to look harder, a simple search for “neural culture” and “vitamin C” or “ascorbic acid” provides plenty of results.

The authors indicated in the discussion (lines 619-623) that “We tested several macromolecular crowders, including Dextran sulfate (DxS500kDa), Poly-sodium 4-styrene sulfonate (PSS200kDa), Polyvinylpyrrolidone (PVP360kDa) for these astrocyte cultures. We ob- served that the largest increase in secretion of collagen IV, fibronectin and laminin (alpha1) was obtained with Ficol40/700.”. Are these results published in this article or elsewhere?

The authors showed that “Our data shows that augmented ECM deposition under MMC+VitC treatments increases the frequency of calcium oscillations, while decreasing the overall synchronicity of the neural network in vitro.” (mentioned in the discussion, lines 651-653), and later relating it to three-dimensionality and complexity of the network. Could you please comment on the biological significance of these results? It would be useful, at least for me.

Materials and methods are very detailed, and the information is most of the times very useful for the reader of the manuscript. But it results sometimes in excessive details, that might result in confusion. As an example, the first paragraph in methods (starting in line 93) seems a bit confusing. Where are the astrocytes derived from, are these the same that are presented later as “iAstros”? Are they cultured in ice cold DMEM at 4ºC? Or are they cultured in “CDI astrocyte culture media”, and is this the same as previously described for astrocyte culture? I would suggest some simpler version, and including a more detailed description if necessary in a supplemental archive, if necessary? The lengthy section also includes some typos and strange constructions, such as the expression “counted in Countess II an automated cell counter” twice repeated. The number of typos and strange punctuation throughout the article is distracting (some examples include “Collage IV” in lines 79 and 308), lack of use of capital letter after period (line 181).

Reviewer 3 Report

This interesting study investigates in vitro, how macromolecular crowding using Ficoll in combination with an enhancer of collagen secretion like L-ascorbic acid, vitamin C, may boost the formation of relevant brain extracellular matrix by cultured human astrocytes. They found that the resultant matrix favors the development of dopaminergic neurons, synaptogenesis, formation of neurites, and calcium activity. Moreover, they observed that macromolecular crowding enhanced the pharmacological responses to compounds pointing to dopamine receptors, thus well emulating what was expected from in vivo data.

Overall, the experimental designs were clear and remarkable, and the paper was written very well. Although the studies are well conducted and the interpretation of data is appropriate, I do have a couple of minor revisions and comments for the investigators to address.

1.      Some typos and grammatical errors need to be corrected carefully. For example, in line 181 there is a full stop in the middle of the sentence “…identify neuronal cells. and for the fixed cell image analysis, we used Hoechst 33342…”. In line 253 there is written “m-opioid receptors” rather than “μ-opioid receptor”. In line 501 “we measured calcium dynamics using single cell fluorescence imaging by expression a genetically encoded GCaMP6f biosensor in the co-culture”: I would rewrite the sentence changing with “…by expression of a genetically encoded…”.

I do recommend carefully checking the manuscript for these typographical errors.

2.      There are repetitions in lines 130,279 and 352. iPSC derived human astrocytes have been abbreviated in line 81 as iAstros, such as Human iPSC derived Dopaminergic neurons (“iDopas”). I suggest using the abbreviation since as been already explained in the first part of the manuscript.

3.      Materials and Methods section. Whether Hoechst does represent a widely use nuclear counterstain, the authors may consider adding some more information about dilution and time of incubation.

Also, the authors should include a brief statement in the “2.4 immunocytochemistry” section, regarding the diluent in which antibodies dilutions have been made.

4.      The resolution and size of the images/graphs are too low. Thus, the graphs are not easily readable for readers to look at the details. They need to be improved.

5.      Results section. In lines 321-323 the authors state “None of the MMC treatments induced large changes in astrocyte numbers and maturity, after 7 days, as shown by GFAP staining (Figure 2f), compared to the control group”. But looking at the graph in Figure 2f, there seems to be an increase in GFAP signal in MMC+VitC group in comparison with the respective control group. There is a mistake in the text/graph or is not clearly explain?

6.      Statistical analysis. I do have some concerns regarding the statistical analysis performed. In lines 268-269 the authors write “The statistical differences were represented as * as p<0.05; ** as p<0.01; *** as p<0.001; **** as p<0.0001 and n.s. as no statistical difference”. However, in the legends of Figure 5 and Figure 6, the p-values are different and is written as follows “Statistical significance: Two-way ANOVA, *p<0.5, **p<0.1, **p<0.01, ***p<0.001, ****p<0.0001, n.s.=not significant”. 

Can the authors explain why the p-values are different? Is this an error or the p-values considered for the statistic are different? Moreover, there is a repetition regarding the statistical symbol ** that is defined as both p<0.1 and p<0.01, which is not possible.

Anyway, if *p<0.5, **p<0.1 have been used, I would recommend repeating the statistical analysis. From a statistical point of view, assuming a statistical significance with a p-value of <0.5 is meaningless.

Reviewer 4 Report

The Authors clearly described an advanced flat co-cultured protocol to enhance the in vitro reliability of neuroglial network physiological responses. 

The manuscript is certainly above average for the quality of the presentation and soundness of the data and thus deserves to be published.

Minor concerns are about citations, indeed authors are referring to their own unpublished work (Kundu et al 2022, 675 Manuscript under review). Moreover. references about the neuronal ECM are from a decade ago or older. I suggest the authors cite more recent papers or reviews also mentioning the role of the neurovascular unit (since they detected lamin and collagen) in the complex CNS plasticity. Lastly, could the Authors add information on why they selected dopaminergic cells as their model?

Round 2

Reviewer 2 Report

The authors addressed all the concerns raised by this reviewer, improving in my understanding the comprenhension of the findings described.

No further comments will be raised from my side.